# A New Framework for the Regeneration Process of Mediterranean Historic City Centres

**Ali Tanrıkul * and Şebnem Hoşkara**

Department of Architecture, Eastern Mediterranean University, 99520 Famagusta, North Cyprus,
Mersin 10, Turkey
* Correspondence: ali.tanrikul@gmail.com

**Abstract:** For thousands of years, cities have evolved with changing needs. Cities are like living organisms, which are exposed to transformations as a result of changing needs and requirements. City centres are one of the attractive, essential, and vital parts of the city that are also affected by these changes. Specifically, historic city centres, which refer to the origins of the city, will be discussed within this context. Urban design aims to shape our cities with better quality and provide better places for everyday life. In addition, urban regeneration can be utilized as generic public policy for solving problems and providing physical improvements for these cities. Although the problems that emerge in each city are similar, sometimes they change circumstantially. As a result, the planning, implementation, and management of urban regeneration projects as well as their sustainability can produce serious complications. This article focuses on the process of urban regeneration, historic city centres, and the Mediterranean region and aims to develop an applicable regeneration framework for historic city centres limited to the Mediterranean region. First, the main problems of these cities are described. Next, characteristics of historic city centres and associated problems of the Mediterranean region are explained. Subsequently, the concept of urban regeneration is clarified, and the processes involved are discussed. Finally, an applicable urban regeneration framework for historic Mediterranean city centres, developed by the authors, is explained with the goal to reduce social segregation while incorporating the contributions of views from both local inhabitants and stakeholders in the process. The methodology of the overall research presented in this article is mainly based on a critical review of primary and secondary documents from the literature through a comparative and exploratory approach.

**Keywords:** sustainability; urban regeneration; regeneration process; framework; historic city centre; Mediterranean

## 1. Introduction

Cities are rapidly changing environments. They are never stable and undulant due to the introduction and adaptation of new circumstances. These rapid changes may affect the nature, economy, social life, and functions of these cities, but mostly negatively through time. In recent years, many cities have been confronted with these negative changes [1–5], which can be summarized as the abandonment of lands and buildings, environmental degradation, unemployment, and damaging social life [4,6–8]. These problems need to be dealt with by utilizing suitable public policies that will return and regenerate life back to derelict land and buildings, create new employment opportunities, improve the environment, and solve problems associated with social life.

Today, rapidly growing technologies play a very important role in city growth. As a result, cities have transformed drastically due to physical and social distortion, structural obsolescence, and loss of function in addition to problems like renewing and preserving the historical urban tissues. More

often historic city centres tend to surrender to these rapid changes; however, by conducting the proper examination and investigations of these historic city centres, healthier spaces and successful attraction points with architectural heritage might be offered to its consumers.

Historic cities with a cultural nucleus, that are still inhabitable, have been either conserved or revitalized. Thus, historic cities are comprised of physical structures as well as tangible and intangible heritages from the past while presenting the culture and the way of living of its people. The importance of conservation and urban regeneration, while planning the future development of historic city centres, may bring life to the community and create vitality by collaborating with the social community and physical context that involves the allocation of novel uses.

Also, cities in Europe and the Middle East that border the Mediterranean have experienced dramatic physical, social and economic changes during the last decades due to relocations, conflicts, and economic crises. These are continents with cities of remarkable histories, cultural inspiration, wealth creation, and social and political dynamism. But during the late 20th century, many cities entered a period of steep decline, losing most of their manufacturing jobs and economic functions, which have resulted in these areas becoming useless, containing dilapidated and abandoned buildings, and developing social and economic problems including redundancy and social deficiency.

The sustainability of historic city centres is vital for cities in providing places of origin, identity, memory, and possession. Based on the arguments above and considering the significance of historic city centres, this study aims to evaluate the components of sustainable city centres concerning these variables. Evaluation of these components and review of the literature suggests that no specific framework for urban regeneration for historic city centres in the Mediterranean exists. Accordingly, the main research question for this study is: 'what factors should be involved in the regeneration process for historic city centres in the Mediterranean to enhance sustainability?'

Therefore, this paper introduces an applicable regeneration framework for historic city centres in the Mediterranean, developed by the authors. The framework discusses the stage processes involved in urban regeneration projects that can be used for development, application, and management aimed at achieving a better urban development standard for historic city centres in the Mediterranean that are in need of regeneration. In the final stage, project control is carried out with the evaluation of all stages, which completes the management cycle of the regeneration process. The aim of the study is to present an applicable regeneration framework for historic city centres in the Mediterranean.

The study presented in this article includes several elements of research obtained through existing literature, and the evaluation of recently published approaches pertaining to the process of urban regeneration of historic city centres in Europe and the Mediterranean. These issues are discussed in the second part of the paper. The backbone of this study consists of the framework that aims to re-define the 'process of urban regeneration' which was developed by the authors while following the "Guidelines for Urban Regeneration in the Mediterranean Region" prepared by the Priority Actions Programme [9], and is discussed in the third part of this paper. Finally, in the last part, using a holistic approach, discussions and future recommendations based on the proposed framework are explored with the aim to set the foundation for the development of future urban regeneration projects for historic city centres in the Mediterranean.

## 2. Conceptual Background: Urban Regeneration of Historic City Centres in the Mediterranean Region

The Mediterranean region is surrounded by the Mediterranean Sea, which is enclosed by Southern Europe, Anatolia, North Africa, and the Levant. Cities of the Mediterranean region have a typical and high-quality character and it is necessary to preserve these attributes, which come from the historical background that follows the genesis and progression of many modern societies.

When taken as a unit, cities are intense, complex, and dynamic structures. A successful city needs the balance of the right mixture of social, civil, residential, and recreational facilities. Also, city centres are the most important habitats that serve the entire population in and outside of the city; however,

city centres have a wide range of needs as well as functions such as accommodation, shopping, leisure, business and transportation, education, and healthcare services. Sustainable cities offer healthy city centres, which may differ from city to city, but set the foundation for outdoor public areas, create a different function zone, provide the quality of the physical environment, offer freedom of movement for vehicle and pedestrians, and most importantly, create a place for people. All city centres operate similarly, which is to be expected from historic city centres as well. While historic city centres provide healthy living environments with their architectural heritage for dwellers, they also need to keep pace with the evolving social and technological changes that occur and which are accompanied by developing urban problems.

Thus, the historical and cultural accumulation of the cities, whose history dates back centuries, is mostly located in the central areas of the cities and their traces can be carried to the present with the aid of historical layers. For this reason, historical city centres are the places where the memories and meanings of the city can be observed and felt most by the reflections of the building stock, urban texture, and traditional life forms. In this respect, the preservation of historical city centres enables the visualization of the historical building stock, keeping the social memory alive, assisting the formation of urban identity, creating a social attraction centre, and providing economic benefits to the city with its cultural centre and touristic potentials. Therefore, it is necessary to protect and refunction these areas in the most accurate way possible. Within the framework of sustainable conservation principles, objectives and aims should be determined for the improvement and development of the historical city centres possessing cultural heritage [10–12]. These values, historic heritage, and local culture, of historic city centres need to be protected and managed by considering the needs of the society living in that particular area. Situations that constitute a threat to historical cities are explained by the Historical Urban Landscape (HUL) approach [13] as follows;

> *"The identity and local character of historic cities are increasingly under threat by the globalizing process of urban development. Traditional heritage conservation practices in the cities have yielded important successes in saving historic areas from urban decay".*

The HUL approach considers local culture and architectural heritage as well as values and meanings as important factors in the decision making process. These issues are recognized through specific steps and tools, such as defining the city's natural, cultural and community resources; consulting with the community and stakeholders; and conducting environmental and social impact assessments, among others [13].

The majority of urban problems primarily occur in unplanned cities as a result of illegal urban expansion. Urban expansion consists of problems such, as the dispersion of urban populations, demographic decline, economic downturn, geographic contraction, and technological revolution, which has led to a migratory shift from the city centre, which has resulted in the shrinkage of these cities [14–18]. Recognition of these problems and taking the necessary precautions are vital steps towards a viable solution. When addressing these problems in these cities, it is recommended that regional and characteristic features be taken into consideration prior to developing proper and sustainable solutions [2,4,5].

Today, most Mediterranean cities have common problems like dense population growth, social disruption, lack of attention towards traditional values, differentiation in revenues caused by change in classes, natural resource abuse, increased air pollution, loss of open spaces, degradation of the ecosystem, increasing land and building values combined with rapid construction, and consequently lack of infrastructure. These problems create physical, social, economic, and environmental damage and become difficult to resolve when timely measures are not taken. The attributes and characteristics of Mediterranean cities can be maintained, preserved, and protected with urban regeneration [2,4,19,20].

It is possible for cities to achieve sustainable results with this kind of an attitude. With the help of theoretical concepts, different approaches have been created for solving similar problems under different conditions. These methods were utilized and applied as a holistic approach to revitalize

and sustain historic city centres. Although there are variations in the unified approaches to the problem rather than be utilized in practice, "urban regeneration" constitutes the focus of this study. As Galdini [21] states;

> "*Various interlinked factors have played a part in this process: the need to breathe life back into and rehabilitate the historic centres of towns and cities, wider-ranging and more diversified cultural pursuits, consumers' interest in the heritage and urban development and their search for things to do and for spending opportunities*".

Urban regeneration is a tool for sustainability. One of the main goals of sustainability is to improve the quality of life and viability of the built, social, and natural environment. As Galdini [2] points out, regeneration encompasses an extensive variety of actions that include renovating the built environment, buildings and infrastructure, re-building premises no longer serving their aimed functions, and bringing back life to formerly dilapidated areas. So, the focus is on upgrading the economic, social, and environmental strength of the city. When the process of regeneration is examined in most European cities, it is obvious that many cities have accepted mutual policies dealing with numerous negative social, economic, and contextual phenomena [2]. The launching process of urban regeneration has two key parameters that play an important role: a strong political will of decision makers and available funds. These aspects serve to increase the possibilities of project initiation. According to Galdini [21], the urban regeneration process, in most European cities aims to act on both economic and urban development.

Urban regeneration is a probability for obtaining sustainable and flexible development, energy efficiency, revised land use, revitalization of old city centres, and the empowerment of citizens [22,23]. Urban regeneration brings conscious, systematized, and planned action and is used for economic, social and physical reconstruction as well as the re-functioning of the physically decayed and old urban areas. This situation is being used to improve existing building areas in physical environments and regulate their functions while greatly contributing to the improvement of the socio-cultural structure within its current economic value.

When the complexity of urban dynamics is taken into consideration, as the purpose of urban regeneration, the primary purpose of urban regeneration according to the Priority Actions Programme [9] is four-fold and includes the following aspects;

- Economic: to facilitate employment, appeal to the interest of stakeholders, rebuild the urban economy;
- Social: to improve the local infrastructure and increase the source of urban housing;
- Environmental: to develop living circumstances;
- Cultural: to attract and enrich architectural heritage and urban tourism.

Fundamental objectives of urban regeneration include reviving the environmental, social, cultural, and economic processes of a city by taking into consideration the complexity of existing urban dynamics and associated complications. Examination of urban circumstances is accepted as the initiation of the process of urban regeneration.

Roberts [4] defines urban regeneration as "*Comprehensive and integrated vision and action which leads to the resolution of urban problems and which seeks to bring about a lasting improvement in the economic, physical, social and environmental condition of an area that has been subject to change*". A similar definition is also offered by Trumbic [5] who suggests that urban regeneration is a generic and combined act and vision aiming to function as a guide in resolving urban causes while finding consistent development in economic, physical, social, and environmental circumstances. Basically, urban regeneration is a process that proceeds with physical change. Urban regeneration suggests that solving circumstances faced in cities follows a long-term implicit process with well-defined determination. It is necessary to provide a strategic view with a long-term plan to realise the purposes of both communal and contextual change. In summary, urban regeneration involves the redevelopment of urban environments experiencing

physical, economic, and environmental degradation; therefore, these valuable opportunities should be taken into consideration so that the urban regeneration process supports urban sustainability [24,25].

By the end of the 1980s, developmental problems were apparent and most Mediterranean based European cities recognized their need for a policy to resolve these issues [26]. To resolve these issues by using the encouragement of remedial activities within a competitive context for both developed and developing economies, one sensible approach and the main objective was to revitalize the city centre and apply ventures to upgrade the quality of the built environment. With this in mind, the city's weaknesses and strengths were considered as determining factors pertaining to the development of policies by local authorities. As stated by Kwon and Yu [3], in order to become and pursue a sustainable and vibrant habitat for citizens, users, stakeholders, entrepreneurs, and visitors, local governments in these cities should develop their own policies.

Results of the current literature review suggest that developed urban regeneration projects aiming at eliminating the problems we observe in today's cities, present designs for protection of the natural environment, social justice, and economic development. Also, taking into consideration the views of people who live in the concerned area from start to project completion will facilitate the implementation of urban regeneration projects more easily.

Including the cooperation of local authorities, civil society, and the private sector in the decision-making process of the central authority will be a beneficial approach for these projects.

As a result, political, social, economic, and environmental requirements must be met to ensure sustainable urban regeneration. Participatory planning, equal income distribution, and increasing the use of renewable resources are issues that need attention.

The basis of planning for the area where the regeneration will take place focuses on obtaining information and analysing this information to shed light on the management of the project. Planning the regeneration project with the information obtained from the field consists of various steps which are carried out according to the planned implementation. In the end, the control is carried out by evaluating all stages in a holistic approach and thus the management cycle of the regeneration process is completed.

Since the information obtained from the literature review is more general and there is no special urban regeneration process that focuses on the historic city centre as a whole, this study intends to fill the gap of redefining a framework that can be utilized for the regeneration process of historic city centres in the Mediterranean. Results from the literature review establish the basis of Section 3.

## 3. Introducing an Applicable Urban Regeneration Framework for Historic City Centres in the Mediterranean Region

### 3.1. Redefining the Process of Urban Regeneration

Urban regeneration is described as an interventionist activity aimed at providing an individual scheme to reflect the needs and requirements of the city and to reduce the social disruption and economic concretion of deficit urban areas [4,27,28]. Roberts [4], on the other hand, stresses the importance of various actors in this activity:

> "*The task of ensuring the effective regeneration of an urban area is of fundamental importance to a wide range of actors and stakeholders, including local communities, city and national government, property owners and investors, economic activities of all kinds, and environmental organisations at all levels from the global to the local*".

The process of urban regeneration describes a collaborative interaction between strategies, objectives, the collaboration and consensus between stakeholders/actors, and the strong adaptation within a city. It is a complex structure and long-term process that includes making an on-site diagnosis, defining the objectives and operational activities, incorporating actors, applying and implementing the project, and managing the project after implementation has been completed.

The most significant result reached at this point is the importance of each stage and dealing with the main issue so that it can be designed using a holistic and multidisciplinary approach. However, this holistic approach is not specialized as a standard framework unique to historic Mediterranean city centres, which is the scope of the study. With this in mind, current practices in the world should be evaluated and suggestions, with basic principles appropriate to the scope of work, should be developed [29–66]. When taking into consideration all of these facts, the review of existing studies suggests that there is need to redefine the process of urban regeneration and provide a new framework that aims to set up a comprehensive and comprehendible evaluation of an applicable framework for Mediterranean historic city centres. As a result, an applicable urban regeneration framework was developed by the authors based on the below mentioned stages.

The first stage of this research is an in-depth review of the literature that was conducted for city centres. The historical development of historic city centres is very significant due to the historical richness of the city as well as its existing artefacts making these city centres a focal and attraction point. These architectural and urban heritages possess considerable values for each city. This is an important objective in sustaining cultural richness but it can be adapted to present conditions and associated with sustainability by local administrations. Within this context, historical city centres have been examined as per the criteria required for sustainability by various scholars [67–72]. Their discourses on the subject have been used to define characteristics, strategies, and developments of historic city centres throughout the historical periods since the first formations of the city.

It is possible for cities, and all of their components, to attain a sustainable structure with urban development policies. So far, a number of cities have produced various approaches to resolve their problems. This solution sometimes encompasses a portion of the city and sometimes recommends the entire city as part of a holistic approach. These suggested solution-based proposals, which could be recommended for either the entire city or just a specific region within the city, comprise an important component regarding the value of the managed approach to be utilized. As there are many approaches that deal with urban development policies, it is important and necessary to understand the concepts of "regeneration", "urban regeneration", and the "process of urban regeneration" to determine the objectives and criteria that are involved as suggested by scholars [1,3–8,21]. In addition, this policy development stage was also discussed in-depth to gain a better understanding of its relationship with the city, city centre, and specifically the historic city centre. At the end of this stage, a critical assessment was done regarding the regeneration of historic city centres on how they can contribute to the sustainability, as part of the developmental process for the framework.

The second stage consists of examining examples of historical cities with urban regeneration project processes, problems, objectives, and policies in European and Middle Eastern cities in the Mediterranean region, by reviewing a number of documents presented in detail. The examples were selected primarily according to their policy approaches and their city centres. The first selection criteria were "historic city centre" and the concept of "regeneration" as an urban policy. Because there is a total of 47 European cities and 17 Middle Eastern Cities within the Mediterranean region, a limitation in city numbers was required for manageability purposes. The selection criteria of "historical city" included having a border along the Mediterranean Sea, having settlements within the historic city centres, having a planned regeneration project and scale of settlement. The regeneration processes of selected historic city centres from Europe; Athens, Greece [29–33], Palermo—Sicily, Italy [34–38], and Valetta, Malta [39–42] and the Middle East; Beirut, Lebanon [43–51], Damascus, Syria [52–60], and Jerusalem, Israel [61–66] were studied. Based on this study, selected examples of cities were marked on the map, prepared by the authors as shown in Figure 1.

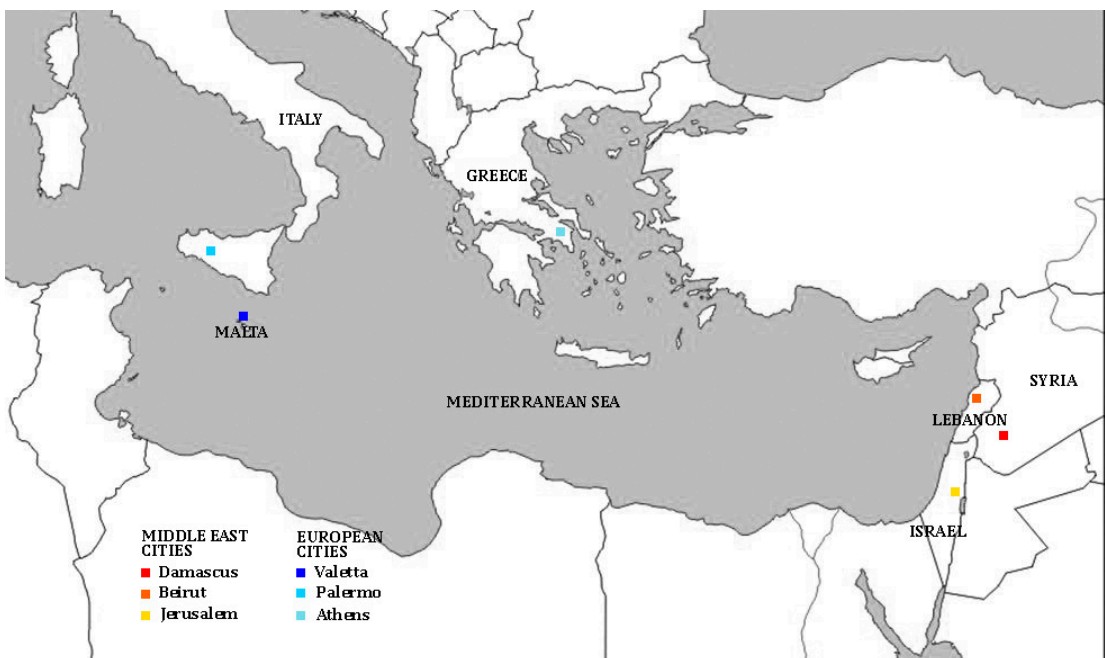

**Figure 1.** The Map of selected examples of cities in the Mediterranean Region; as developed by the authors.

Taking into consideration the abovementioned historic city centre examples, factors that may have contributed to successful strategies, the methods and procedures, the resources utilized, and the obtained results were examined. The evaluative and explorative methodology used for these examples were categorized as follows: city name, area location and characteristics, urban complications (social-economical-physical), area possibilities, strategic objectives of the regeneration project, limitations of the project, programming (phasing and timing) of the regeneration project, key stakeholder identification, research for development results, and achieved outcomes. This detailed structure provides the main framework for the examination of larger samples. With this classification and identification methodology, the possibility of selecting a suitable urban regeneration project, its applications, and notional point of views are identified. The structure in this section allows for the comparison and evaluation of examples as well as the discovery of similarities and differences in urban regeneration projects and policies for each city. Many important and useful results can be drawn after this comparison and assessment. This system also provides an overview and outline of existing urban regeneration policies, which can easily benefit in terms of policies, programmes, and objectives. Also, reviewing the criteria for the selected examples offers the possibility for inter-city comparisons. As a result of this stage, findings are concluded based on the lessons learned from the examples, which are crucial in framework development.

The third stage regarding the process of urban regeneration includes extensive readings and analyses of the comprehensive guideline for urban regeneration—"Guidelines for Urban Regeneration in the Mediterranean Region" prepared by the Priority Actions Programme (PAP). The "Guidelines for Urban Regeneration in the Mediterranean Region" considers urban issues in the Mediterranean region while addressing urban regeneration as a remedial tool. The PAP, a subgroup of the Mediterranean Commission for Sustainable Development (MSCD), analysed existing problems in selected Mediterranean cities and compiled detailed guidelines in this report. The objective of this report was to offer advice regarding the implementation of urban regeneration process tools; however, these tools were not given in a step-by-step fashion for users. At this point, due to the availability of the sources and the scope of the report, it was selected as a main source for the development of the framework developed by the authors; however, in a more user friendly format with easy to follow schematized steps.

The fourth stage consists of merging the first three stages together into a unified framework that aims to redefine the urban regeneration process, which is based on existing literature, selected examples, and analysis of PAP's guidelines that are currently in use. The goal of this stage was to provide a working model to simplify the process of urban regeneration while maintaining specified professional standards.

The final stage consists of transferring the proposed processes into a figure format and explaining each process on a step-by-step basis to suggest an applicable urban regeneration framework for historic city centres in the Mediterranean region. The step-by-step frameworks and flow diagrams can be found in detail in the next sections.

### 3.2. Development of an Applicable Framework for Historic City Centres in the Mediterranean Region

The proposed framework is used as a measure for urban regeneration projects in historic city centres, where urban regeneration is to be implemented. The proposed urban regeneration process was schematized and explained in a step-by-step fashion to make it easier to follow and understand. This part of the study summarizes the steps that urban regeneration projects should follow during the implementation process. Based on the review of literature, case studies, and the guidelines for providing an applicable urban regeneration process framework, the planning process of urban regeneration is divided into four main steps: the starting step, the launching step, the planning step, and the implementation and management step. In addition, these steps are divided into sub-steps, which will be used as a checklist to question the appropriateness of projects. Each of these steps and sub-steps has their own priorities and associations. At each step, there are distinctive tools that local governments can use to systematically design a regeneration process.

### 3.2.1. The Starting Step: "Kick-Off"

One of the first steps for setting up the process of an urban regeneration project is the starting step which begins with analysing the existing urban problems, scope of scale, and stimuli of a process. In considering a city's past, analysing urban problems is a critical necessity required in the prediction of its future development. The unique structure of the city is analysed to provide insights into the project on the positive qualities of the city. Moreover, urban problem analyses identify these problematic areas. Urban problem analyses of historic cities include understanding the built and natural environment as well as its cultural aspects, which gives the city its unique character. Based on the reviewed examples [29–66] in each particular case the scope of scale illustrates the urban regeneration project sizes, which can be classified as small, medium, or large scale, and determining the scope of the urban regeneration project to be implemented is based on the context of the historic city. The starting step provides a class of stimuli of the project. Stimuli selection can be affected by each case's particularities, which can be categorized as economic, social, environmental, or institutional.

Using the outcome data, the process of urban regeneration needs to clarify that "*the particular complexity of urban problems in each city drives to a large extent the need for urban regeneration, its focus and scale*" [9]. With this in mind, it can be suggested that;

-    The goals and objectives, which refer to the need, are the required end for the urban regeneration process.
-    Various multiple-dimension actions in the form of key interventions/projects can be supplied by the focus for the urban regeneration process.
-    The spatial and financial extent of the intervention presents the scale in the suggested framework for the urban regeneration process.

At the end of this step, some key features are provided which are long term perspectives, political will and commitment, multi-actor/stakeholder participation, organizational frameworks, institutional/legal frameworks, and financing and maintaining the process. These key features are the key trigger of the project process as the final sub-step before continuing onto the second step of

the process [9]. Based on this analysis, the Starting Step of the Framework for the Process of Urban Regeneration is redefined by the authors in Figure 2.

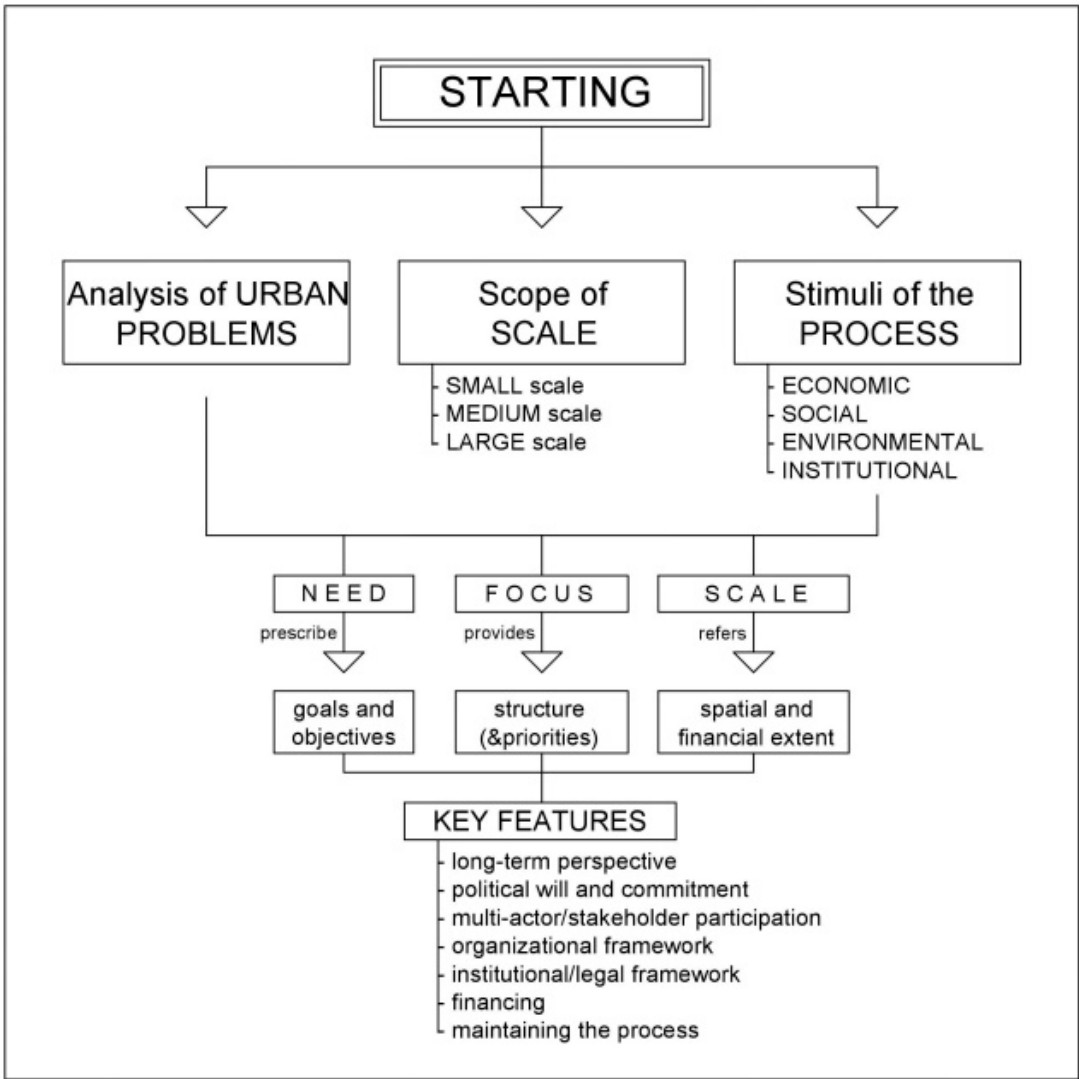

**Figure 2.** The starting step of the framework for the Process of Urban Regeneration; as developed by the authors.

### 3.2.2. The Launching Step: "Taking Action"

While the starting step provides key features for the regeneration project, the launching step identifies the actions that need be taken. The launching step of the regeneration process starts with the diagnosis of the basic problems and builds up a framework to identify project actions required for the planning step. Therefore, diagnosing the basic problems is supportive in defining the restrictions and presences of each individual spot as well as the related prospects needed for regeneration. This should start with recognizing the existing situation, exploring trends, and mapping the existing problems and opportunities [9]. To build up an urban regeneration process framework for historic cities, the existing institutional structure can provide a good basis. This structure is used to define the necessities for institutional action which are governmental, legislative, financial management and financing, promotion, and communication and participation; however, it is also important to identify the key actors from the civic sector who are experts (urban designers, infrastructure, transport and environmental planners, sociologists, and economists), private entrepreneurs, NGO's, resident community, and international agents (EU, UN, etc.). This step of the process ends with identifying

project actions that include goals, objectives, and characteristics. This process, defined by the authors, is shown in Figure 3.

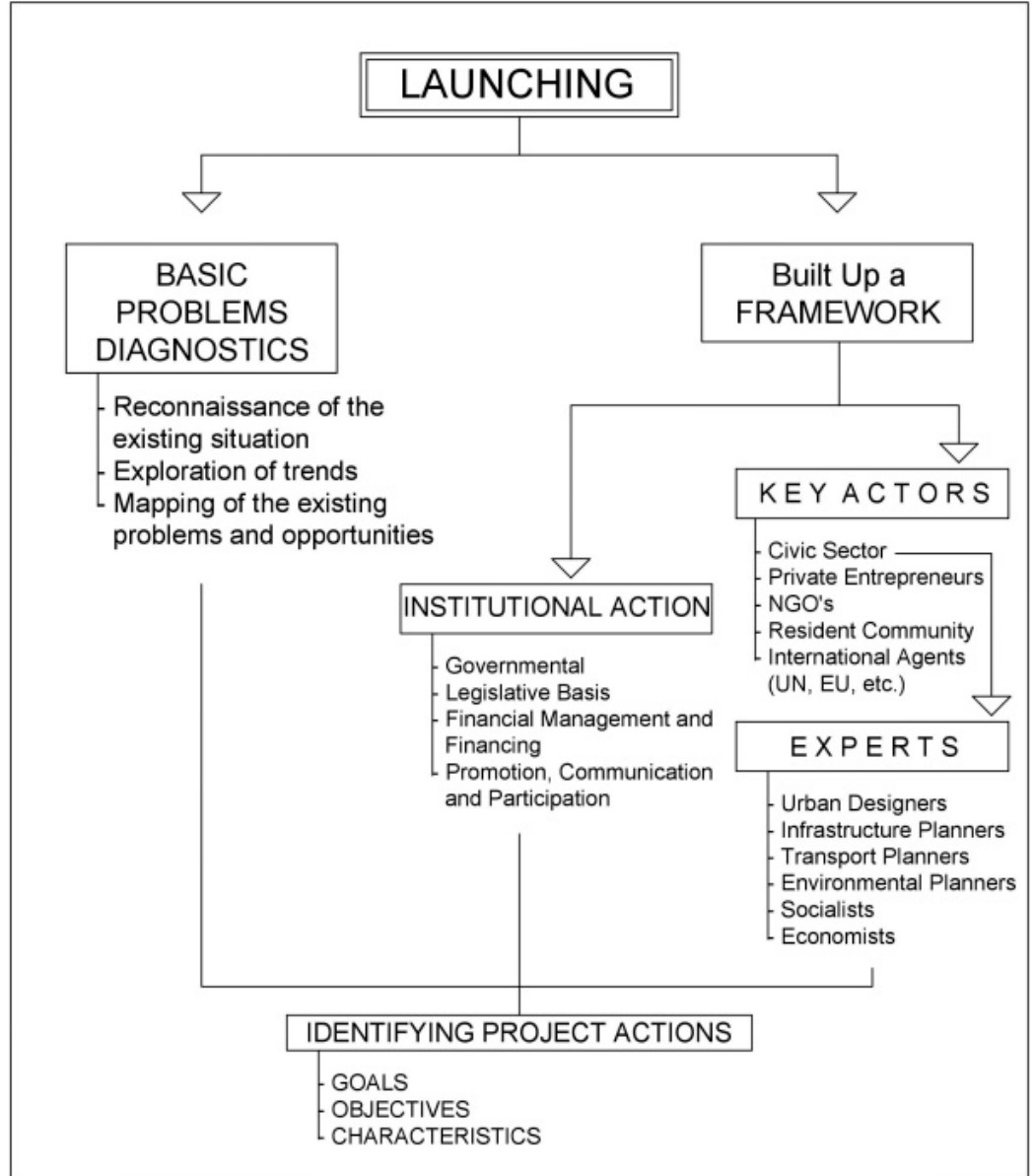

**Figure 3.** The launching step of the framework for the Process of Urban Regeneration; as developed by the authors.

### 3.2.3. The Planning Step: "Deliberation"

The planning step starts with analysing the current situation. The most pressing issues of concern, for the current situation, are squatter areas, old town districts, historical centres, existence and formation of urban derelict land, social unease and unrest, economic decline of city centres, and visual disarray and disorganised spaces around the entrance to the city. Understanding the current situation must include an in-depth examination of the existing structure and the state of the land. The planning step provides a developing strategy, which clarifies the goals and objectives as well as actions necessary for urban regeneration, which are vital components of the third step of the process. In the goals and objectives stated in the Guidelines for Urban Regeneration in the Mediterranean Region, "the key planning interventions", "the financial scheme" and "organizational structures" turn into a strategy that provide the basic project characteristics [9]. The main goals could include the rehabilitation of complex

urban structures, improvement of the environment (quality of life), preservation of valuable and unique fabrics, control the deterioration of specific urban zones, and restructuring economic activities of the regeneration projects [9]. The objectives could also include restoring buildings, rehabilitating private residences and upgrading the infrastructure, identifying business needs, developing new opportunities, encouraging economic growth, organizing institutions for management and planning, enhancing basic services, empowering communities, and promoting public participation [9]. Therefore, urban regeneration strategies are location specific and this determines the tools and instruments that should be used. In this regard, the scope of operations can be defined as the plot, the block or the district, where the action of the process will take place as a sub-step. This step of the process ends with the implementation of the plan, which is the most significant part of carrying out the process of urban regeneration. Factors involved in the plan implementation include land control, traditional tool selection for spatial planning, project-based strategic planning, specific plans, programming, flagship developments, and implementation stages. Figure 4 shows that the Planning Step of the Framework for the Process of Urban Regeneration, as defined by the authors.

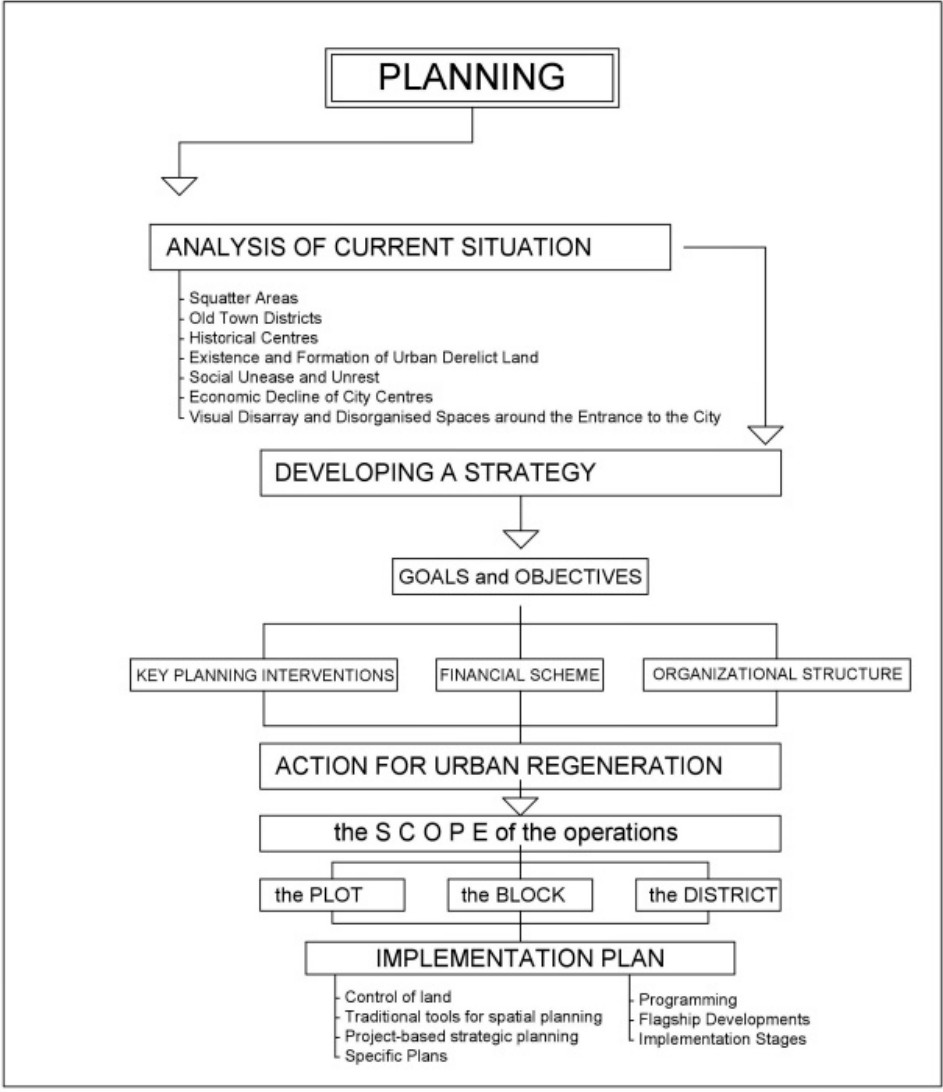

**Figure 4.** The planning step of the framework for the Process of Urban Regeneration; as developed by the authors.

### 3.2.4. The Implementation and Management Step: "Decipher"

The last and most important step of the process is implementation and management, which depends on many factors including public participation and partnership, funding, governance and participation, and monitoring and evaluation. Different actors take part in the management process throughout its operation, the allocation of financial resources and funds, monitoring, evaluation, employing technical proficiency, and arranging basic organizational activities which are essential components for such a multifaceted action and have been detailed in each sub-step. Public participation is a key contribution of different actors who are public sector agencies, local authorities, regional and national agencies, residents, local community, NGO's, government agencies, and community-based groups for the urban regeneration process and multiple level partnerships could provide a clear picture of a city's characteristics, problems, prospects, and needs while forming a shared vision for the city, identifying common needs and priorities pertaining to urban regeneration, promoting a commitment to implementing the project, and generating the necessary funding for project implementation as well as following up with urban regeneration projects. Funding is a crucial requirement for urban regeneration projects and requires the support of private partners, local and central authorities, and international organisations. Governance and participation are recognized as essential features for successful implementation of urban regeneration projects. As monitoring and evaluation are also key components of any urban regeneration project, the Guidelines for Urban Regeneration in the Mediterranean Region [9] suggest that providing project sustainability places fundamental emphasis on having suitable specialists and well-equipped manpower throughout the entire process. The Implementation and Management Step of the Framework for the Process of Urban Regeneration is defined and presented in Figure 5.

The final product of this process, which can be considered for each regeneration project targeting historic city centres, could be different for each city and it is important to note that each result will be specific to the field it is applied to. While the data collection phase, which includes analysing the current situation in the area of regeneration and identifying the stakeholders, provides detailed data for the field, information obtained from each project should be examined in detail so that the level of functionality for the entire process can be seen.

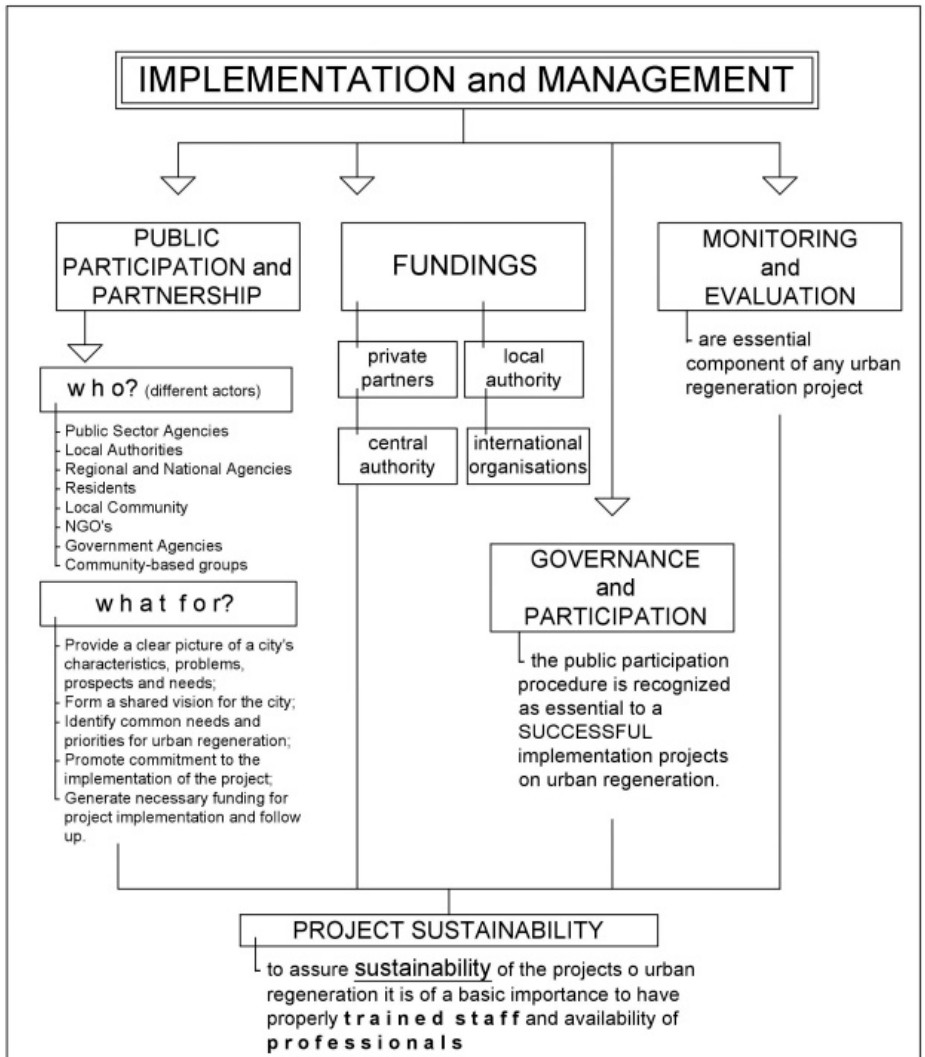

**Figure 5.** The implementation and management step of the framework for the Process of Urban Regeneration; as developed by the authors.

## 4. Conclusions and Future Recommendations

Nowadays cities are obstacles with difficulties for various reasons; however, despite this, people should have equal access to housing, job opportunities, education, and health care, which are among the growing problems of growing urban populations. For this purpose, the main concern of this study is to develop a proposal for creating an operational standard that can be used in the development of people-oriented and sustainable urban regeneration projects, to reach universal design standards in urban and building scales, to prevent the destruction of nature, and to produce solutions for the benefit of all stakeholders from an economic point of view. The aim of this study was to present an applicable framework pertaining to the urban regeneration process of historic city centres in the Mediterranean region. The suggested urban regeneration framework is promising for various actors ranging from urban designers to local authorities, as well as citizens.

In this study, the development of urban design and the processes involved was explained based on the existing literature. The key outcome from this review suggests that each stage involved is crucial and that a multidisciplinary approach must be used for healthy and sustainable outcomes. However, current practices utilized worldwide should be examined, evaluated, and recommendations based on these observations should be developed and incorporated into the abovementioned principles, which in turn can be applied to local conditions. Taking into account the basic principles identified for each

area, where the transformation will be carried out, attention and caution must be taken into account prior to implementing any application and these implementations must be designed according to the basic needs as well as the economic and social structure of the area.

Urban regeneration project adaptation will be attained much more easily when project development and designs aimed at eliminating the problems we face in cities such as protecting the natural environment, establishing social justice and economic development, and inhabitant rights are taken into consideration. Therefore, the political, social, economic, and environmental requirements must be met to ensure sustainable urban regeneration. Participatory planning, equal income distribution, and increasing the use of renewable resources are among the main issues that need to be taken into consideration as well.

The management of the process of urban regeneration, which the authors have attempted to define within the context of the study, provides a way of accessing the points where urban transformation works are desired. It is not possible to actualize healthy urban transformations without evaluating today's social, economic, environmental, and physical criteria.

This study aimed to develop an applicable urban regeneration framework that could be used for the urban regeneration process aiming for sustainable planning on a step-by-step basis. Depending on the implementation of the proposed applicable urban regeneration framework, some basic standards were targeted. Figure 6 presents the overall step-by-step process of the urban regeneration process. Each step and sub-step is shown in detail and a complete cycle framework was created by linking the transitions between the steps.

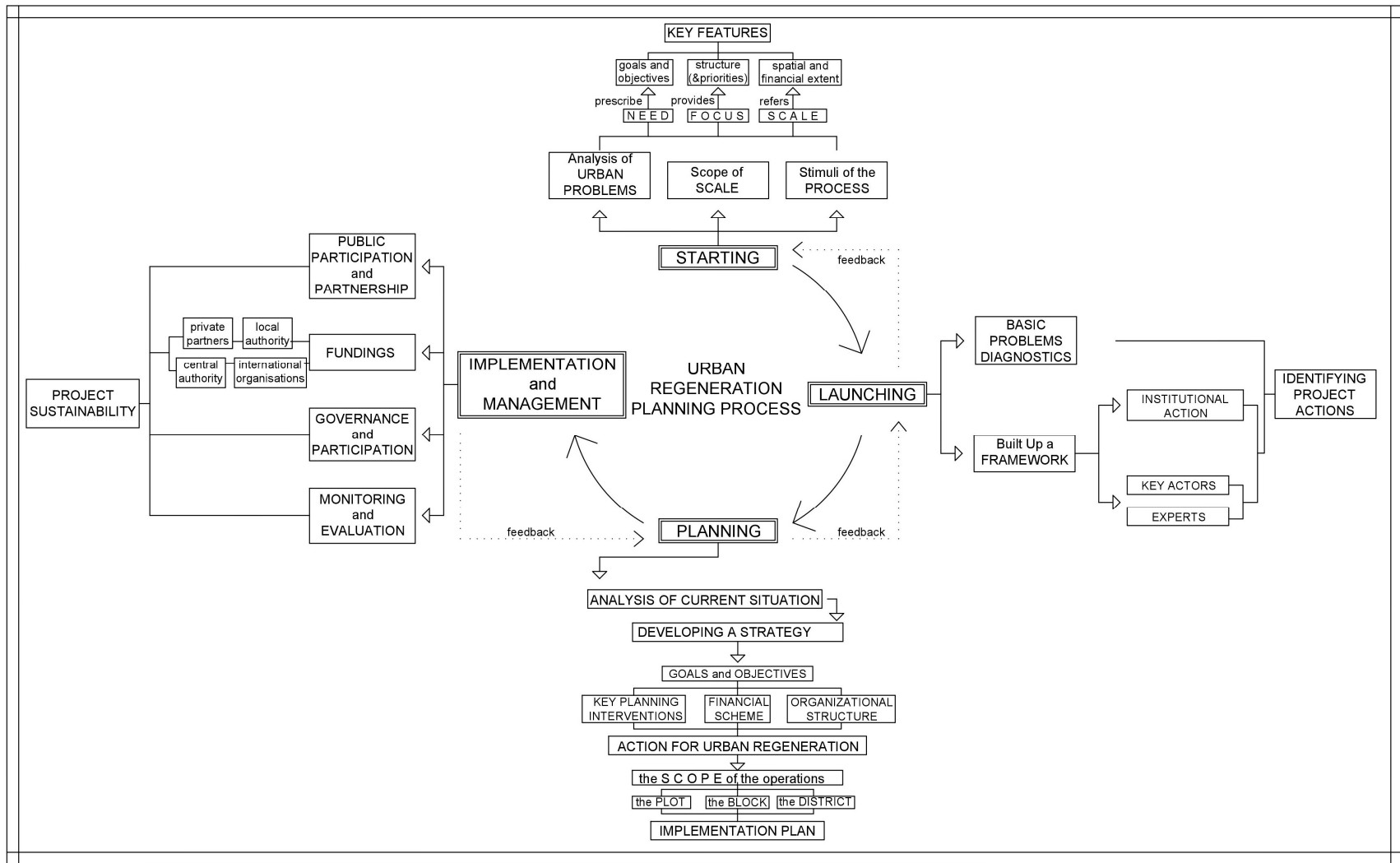

**Figure 6.** The cycle of urban regeneration planning process; as developed by the authors.

In the final stage of urban regeneration, a complete process cycle of the framework was developed. The cycle begins with the starting step of "kick-off", which leads to the launching step of "taking action", which encompasses all actions needed, then continues with "deliberation", which defines the planning steps of the implementations that will be taken, and ends with "decipher", which is the implementation and management stage. It should also be clarified that any stage might require feedback from the promoted experiences as a part of the proposed framework. The most important highlight from this study is that the urban regeneration process is portrayed as the result of a combined approach where, if the right steps are followed, a balance between physical, social, economic, and environmental issues can easily be achieved. The framework presented in this study facilitated the creation and follow-up of these steps.

This framework is prepared for professionals from different stages of governmental bodies, institutions, and experts. Likewise, the components that were clarified during the literature review and analysis were integrated into the framework, as these are the actual results of authorities who are currently facing planning problems. Therefore, the framework covers all of the steps involved in the regeneration process. Concisely, it will be more applicable for the sustainability of urban regeneration projects for historic city centres in the Mediterranean.

The authors believe that the results from this study offer better insight regarding the application of innovative interferences that can be used to facilitate regular connections with inhabitants and consumers, better organization and implementation, and more effective and specialized ways to achieve tasks associated with urban regeneration. The cycle of the urban regeneration process set forth in this study could be an effective and applicable framework that can be followed during the design, implementation, and evaluation processes of urban regeneration projects, especially for the historic city centres located throughout the Mediterranean. While this study emphasizes the development of a proposal for future applications of urban regeneration it also develops a platform for new research to be conducted within the proposed framework. In addition, this study also provides a usable background for evaluating the data that may arise during the evaluation and documentation of future works. As the scope of this study provides an opportunity to create a new multifaceted framework for rapid data collection, analysis, and evaluation for future projects, some of the research recommendations presented in this work lie outside the traditional research approach. Further applications and testing of this framework are needed and can be conducted in the other Mediterranean cities as the proposed framework could provide a baseline approach for further regeneration projects for other cities characteristically different from the Mediterranean region.

**Author Contributions:** The research was designed by A.T. in collaboration with co-author, Ş.H. The first and final drafts were written by A.T. The improvements suggested by the reviewers were implemented by both authors. All authors read and approved the final manuscript.

**Funding:** This research received no external funding.

**Conflicts of Interest:** The authors declare no conflict of interest.

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
