# Peer review of "A New Framework for the Regeneration Process of Mediterranean Historic City Centres"

_sustainability, doi:10.3390/su11164483_

Round 1

Reviewer 1 Report

The work concerns a very interesting problem, but it is characterized by a small degree of detail. The authors make a lot of general statements, which in my opinion need to be explained.

lines 105-112: in my opinion, a reference to literature is needed

lines 187-189: "The review of existing studies suggests that there is need to redefine the process of urban regeneration and provide a new framework that aims to set up a comprehensive and comprehendible evaluation of an applicable framework for Mediterranean historic city centres." – Please explain why such a conclusion was drawn from the literature research carried out?

lines 191-192: "The following items are under debate in a separate comprehensive study, which was taken from an ongoing PhD study of the corresponding author." – Is it partly published somewhere? If so, it is better to refer to these publications. If not, I see no reason to mention it. The reader may be interested in it and it is not clear from such a sentence where and when he will be able to read about it.

lines 216-218: "The second stage consists of a comprehensive literature review containing historical city examples with urban regeneration project processes, problems, objectives, and policies in European and Middle Eastern cities in the Mediterranean region." – Why is there no reference to this research in the article?

lines 220-222: "Because there are a total of 47 European cities and 17 Middle Eastern Cities within the Mediterranean region, a limitation in city numbers was required for manageability purposes." - How many cities were finally selected for analysis and which cities are concerned? A map with marked cities that met the criteria of historical cities or at least a list of these cities could be useful.

lines 283-286: "The scope of a scale illustrates the urban regeneration project sizes, which can be classified as small scale, medium scale, or large scale, which may vary according to the context of historic city, to determine the scope of urban regeneration projects that can be implemented." How many of these projects were there? How many of these projects were there in particular cities? What were the projects?

Można by dodać sekcję "Metodologia i materiały", w której oprócz brakującego opisu metodyki badawczej udzielono by odpowiedzi na pytania dotyczące materiałów badawczych (uwagi w wierszach 220-222 i 283-286).

Section 3.2 describes the Framework for the Regeneration Process of Mediterranean Historic City Centres, but it is still unclear what the results of the literature research that formed the basis for establishing this framework are. In my view, there is a lack of a 'Results and Discussion' section where the results of literature research and project analysis should be described. It is only after that that that one can propose one's own solutions.

In addition:

A small number of literature items, and those that are given do not refer to the latest research. The latest quoted article dates back to 2011. I propose to expand the literary research and add the publications from the last few years. It will certainly partly help to improve the text resulting from the remarks on lines 105-112, 191-192 and 216-218.

In the PDF version, the drawings are not easy to read. The diagrams should be enlarged so that they cover the width of the entire page. Figure 5 could be placed on a page with horizontal orientation.

Author Response

Thank you.

Reviewer 2 Report

I believe this is a good solid article and interesting. I come from a heritage perspective, so it was helpful for me to see this from a planning point of view. I think that there should be some reference to the Historic Urban Landscape (HUL) approach, proposed by ICCROM and applied in the Naples area of Italy. (https://whc.unesco.org/en/hul/) The HUL proposes a combined built and natural environmental approach to the analysis of cities and their surroundings. I also think some reference to the Integrated Urban Conservation ideas of Italian theorists in the 1970s and 1980s would be useful. They were interested in issues of class and gentrification and wanted to address economic goals as a part of heritage management. Key texts would be Giovanni Astengo on Assisi, Giuseppe Venuti and Pierluigi Cervellati on Bologna in 1969, Saverio Murati and Venice in the 1960s, Gianfranco Caniggia, Rome in the 1970s, and Leonardo Benevelo, and the Borgo, Rome, 2004. Overall, I thought the references a little weak. A few more concrete examples of cities would be helpful too.   

Author Response

Thank you.

Reviewer 3 Report

The subject of the paper is very interesting for urban planning disciplines: the Regeneration Process of Mediterranean Historic City Centres. 

The goal of the paper is very ambitious: it aims to develop an applicable regeneration framework for historic city centres in the Mediterranean region.

The structure of the paper is clear but it is treated in a simple way, not considering numerous peculiarities of historic centres such as their role of cultural heritage for city's population (see for ex. Rotondo F, Selicato F, Marin V, Lopez Galdeano J (eds.) (2016). Cultural Territorial Systems. Landscape and Cultural Heritage as a Key to Sustainable and Local Development in Eastern Europe. Springer) or shrinking phenomena  (see for ex. Elke Beyer, Anke Hagemann, Tim Rieniets, Philipp Oswalt (2006), Atlas of Shrinking Cities, Hatje Cantz), which can modify in a relevant way urban regeneration process, strategies and goals.

it's true that Mediterranean Historic City Centres share similarities but also differences so a regeneration framework, if we can think that can be built, needs to be flexible enough to include problems and solutions.

In the phase of implementation and management, in my opinion, it needs to be considered also possible feedbacks by the experiences promoted.

Author Response

Thank you.

Round 2

Reviewer 1 Report

I am very sorry for one comment in Polish. I accept all changes. This version is much better. Thank you for your cooperation.

Reviewer 3 Report

The second review has followed the suggestions sent.

The paper can be published, in my opinion.